# Balancing act: Unraveling the link between muscle strength, proprioception, and stability in unilateral hip osteoarthritis

**Batool Abdulelah Alkhamis**[1], **Ravi Shankar Reddy**[1]*, **Khalid A. Alahmari**[1], **Mastour Saeed Alshahrani**[1], **Ghada Mohammed Koura**[1], **Olfat Ibrahim Ali**[2], **Debjani Mukherjee**[1], **Basant Hamdy Elrefaey**[1]

1 Department of Medical Rehabilitation Sciences, College of Applied Medical Sciences, King Khalid University, Abha, Saudi Arabia, 2 Physical Therapy Program, Batterjee Medical College, Jeddah, Saudi Arabia

* rshankar@kku.edu.sa

**Data Availability Statement:** All relevant data are within the manuscript and its Supporting Information files.

## Abstract

The objectives of this study are to compare hip muscle strength, hip joint proprioception, and functional balance between individuals with unilateral hip OA and asymptomatic individuals and to examine the relationships among these variables in the hip OA population. In a prospective cross-sectional study, 122 participants (unilateral Hip OA: n = 56, asymptomatic: n = 56) were assessed at the CAMS/KKU musculoskeletal Physical Therapy laboratory. Ethical standards were upheld throughout the research, with informed consent obtained. Hip muscle strength was measured using a hand-held dynamometer, hip joint proprioception with a digital inclinometer, and functional balance using the Berg Balance Scale (BBS) and Timed Up and Go (TUG) test. Hip OA individuals exhibited significantly lower muscle strength and proprioceptive accuracy, and poorer functional balance than controls (p < 0.003). Correlation analyses revealed a positive correlation between muscle strength and BBS scores (r = 0.38 to 0.42) and a negative correlation with TUG test times (r = -0.36 to -0.41). Hip joint reposition sense (JRS) in flexion showed a negative correlation with balance (r = -0.46), while JRS in abduction was positively correlated (r = 0.46). The study highlights the clinical importance of muscle strength and proprioception in functional balance among individuals with unilateral hip OA. The results support the incorporation of muscle strengthening and proprioceptive training in interventions to improve balance and mobility in this population.

## 1. Introduction

Osteoarthritis (OA) of the hip is a prevalent degenerative joint disease that significantly impacts the musculoskeletal system, leading to pain, disability, and a diminished quality of life [1, 2]. The pathophysiology of hip OA involves not only the cartilaginous wear and changes in bone structure but also extends to muscular weakness, proprioceptive deficits, and balance

**Funding:** This research was funded by King Khalid University, grant number: GRP/193/44.

**Competing interests:** The authors have declared that no competing interests exist.

impairments [3, 4]. These factors contribute to the complexity of the condition and the challenge it poses to clinical management [4].

Muscle weakness, particularly in the hip region, is a well-documented sequela of OA [5, 6]. The hip musculature plays a crucial role in joint stabilization and movement [7]. When afflicted by OA, the hip flexors, extensors, and abductors often exhibit reduced strength, which can be a consequence of disuse, pain-related inhibition, or neuromuscular alterations [8, 9]. This muscular impairment has direct functional repercussions, as it is closely related to an individual's ability to perform daily activities and maintain postural equilibrium [9].

Proprioception, the sensory ability to perceive joint position and movement, is another critical aspect affected by hip OA [10]. Proprioceptive information is integral for the central nervous system to produce appropriate motor responses for joint stabilization and coordinated movements [11]. In hip OA, the deterioration of joint mechanoreceptors alters proprioceptive feedback, leading to a decline in joint reposition sense (JRS), which may further compromise functional balance and increase the risk of falls [4, 12, 13]. Functional balance, the capacity to maintain a base of support with minimal postural sway, is essential for performing both static and dynamic activities [14]. It is a complex attribute that relies on the integration of muscular strength, proprioceptive input, visual cues, and vestibular function [15]. Hip OA patients often experience balance disturbances, which are suggested to result from the interplay between muscular and proprioceptive deficits [16, 17].

Despite the recognition of these issues, there remains a gap in the comprehensive understanding of how these factors—muscle strength, proprioception, and functional balance—interrelate, especially in unilateral hip OA [18]. This understanding is crucial for developing targeted therapeutic strategies [13]. Prior research has individually explored these aspects, but an integrated approach is lacking. Therefore, this study aims to bridge this knowledge gap by employing a cross-sectional design to compare these critical factors between individuals with unilateral hip OA and asymptomatic individuals. In doing so, the study endeavors to elucidate the potential relationships between hip muscle strength, joint proprioception, and functional balance, which could inform the tailoring of rehabilitation interventions to improve patient outcomes in hip OA. A comprehensive investigation into the musculoskeletal consequences of unilateral hip OA and the interdependencies among key factors affecting patient mobility [19, 20]. The findings of this study could significantly influence clinical practices, offering new insights into the management and treatment of this debilitating condition [20].

In summary, the primary objectives of this study are to compare the levels of hip muscle strength, hip joint proprioception, and functional balance between individuals with unilateral hip osteoarthritis and asymptomatic control subjects, and to examine the interrelationships among these variables within the hip OA cohort. We hypothesize that individuals with unilateral hip OA will demonstrate significant deficits in muscle strength and proprioception when compared to asymptomatic individuals and that these deficits will be associated with decreased functional balance. Additionally, we anticipate that the strength of the relationships among these variables will provide empirical support for the development of targeted rehabilitation strategies. This research aims to contribute to a more nuanced understanding of unilateral hip OA and its impact on musculoskeletal function, which is essential for crafting effective interventions to enhance the quality of life for affected individuals.

## 2. Materials and methods

### 2.1 Research design and ethics

This study utilized a prospective cross-sectional research design, conducted between 5th April 2021 and 20th February 2022, at the CAMS/KKU Musculoskeletal Physical Therapy

Laboratory. The laboratory is fully equipped to conduct musculoskeletal assessments and balance testing. Ethical approval for the study was obtained from the DSR and KKU Institutional Review Board (protocol code: ECM#2021–4504 and date of approval: 10-03-2021), and all procedures strictly adhered to the ethical guidelines outlined in the Declaration of Helsinki. Written informed consent was diligently obtained from all participants, who were comprehensively briefed about the study's objectives, and procedures.

## 2.2 Participants

A total of 122 participants were recruited for this study. Two groups were formed: individuals diagnosed with unilateral hip osteoarthritis (n = 56) and asymptomatic individuals (n = 56) without any hip joint pathology. The inclusion criteria for the hip osteoarthritis group encompassed several key requirements. First, participants needed to have a clinical diagnosis of unilateral hip osteoarthritis, which was determined through a combination of radiological and clinical assessments. Additionally, they were required to meet symptomatic criteria in line with the American College of Rheumatology's Clinical Criteria for Classification and Reporting of Hip Osteoarthritis [21]. Furthermore, participants had to report experiencing a minimum of 30 mm of pain on a 100 mm scale while walking, which corresponded to the initial question in the hip-specific Western Ontario and McMaster Universities Arthritis Index (WOMAC) [22]. Age was also a factor, with individuals falling within the age range of 40 to 75 years being eligible. Moreover, participants needed to demonstrate their ability to ambulate independently, whether with or without the use of assistive devices. Lastly, individuals with a history of hip joint surgery within the previous six months were excluded from the study based on these criteria. Inclusion criteria for the asymptomatic group included the absence of hip pain or discomfort and age-matched to the hip osteoarthritis group. Similar ambulatory capabilities.

The exclusion criteria for participation were: any contraindications to muscle strength or proprioception assessment, presence of lower limb joint disorders (excluding hip OA), history of surgery impacting balance or muscle strength, and musculoskeletal or neurological conditions affecting hip function. Asymptomatic participants were matched to the OA patients based on the side of dominance. For patients with right-sided hip OA, we matched the data from the right (dominant) side of the control participants. This was reversed for patients with left-sided hip OA. We standardized the assessment procedure across all participants to minimize side-to-side variability.

## 2.3 Sample size calculation

The sample size calculation was focused on the primary objective of comparing hip muscle strength, hip joint proprioception, and functional balance between the two groups. Using G*Power software, we determined that a minimum of 56 participants per group would be necessary to detect significant differences in the primary outcomes. This calculation was grounded on an effect size of 0.4 [13], an alpha level of 0.05, and a power of 0.8. Thus, the total sample size required for the study was 112 participants, and the effect size used in this calculation was derived from a prior study conducted by Freddy et al [13].

## 2.4 Hip muscle strength testing using hand-held dynamometer

In the assessment of hip muscle strength using a hand-held dynamometer (MicroFET 2 –Hoggan Health), three specific muscle groups were targeted, namely the hip flexors, extensors, and abductors. To ensure consistency and reliability, participants were meticulously positioned following standardized procedures for each muscle group. This positioning was critical to isolate

the specific muscle groups and optimize the accuracy of the measurements. During the testing procedure, three consecutive measurements were taken for each of the targeted muscle groups. This triad of measurements aimed to capture a reliable representation of the participants' muscle strength capabilities. The maximum force output generated during each of these measurements was quantified in Newtons, providing a precise and objective measurement of muscle strength for the hip flexors, extensors, and abductors.

In the evaluation of hip flexion strength, the individual undergoing testing was positioned in a seated posture, with the hip flexed at an angle of 90 degrees. Both upper extremities were employed to securely grip the sides of the examination table to ensure stability. Analogous to the preceding assessments, the examiner applied resistance at a fixed position, and the individual undergoing testing exerted a maximal effort against the dynamometer while effectively opposing the examiner's resistance. Resistance was consistently administered approximately 5 cm proximal to the proximal edge of the patella, with a specific focus on targeting hip flexion (Fig 1). In the evaluation of hip abduction strength, the subject undergoing assessment adopted a side-lying position, ensuring that the tested hip remained in a neutral alignment while the contralateral hip was flexed at an approximate angle of 90 degrees. To establish stability, the participant grasped the side of the examination table with their upper extremity and positioned their head on the lower arm. Pelvic stabilization was meticulously provided by the examiner using one hand, while the other hand applied resistance at a fixed location. The participant was explicitly instructed to exert a maximal force against the dynamometer. The resistance was consistently administered approximately 5 cm proximal to the flexed knee, with a specific emphasis on targeting hip abduction (Fig 2).

In the assessment of hip extension strength, the individual assumed a prone position, with the targeted hip maintaining a neutral alignment, while the knee was flexed to an angle ranging from 70 to 90 degrees. Both upper extremities of the person being tested were employed to

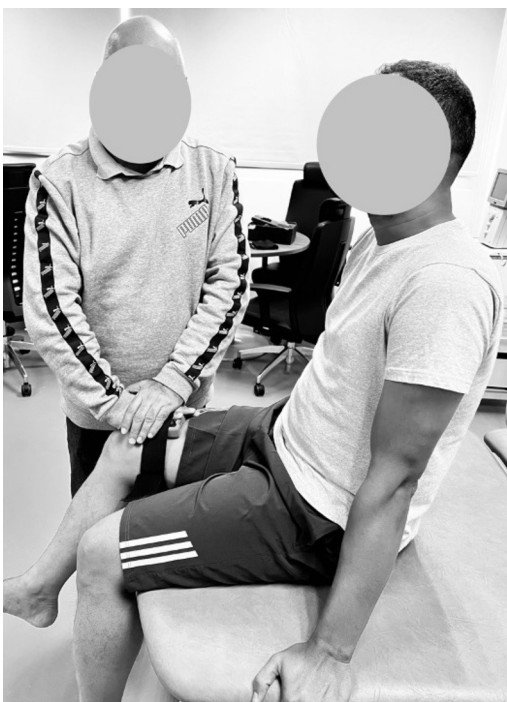

**Fig 1. Hip flexor isometric strength assessment using a digital hand-held dynamometer.**

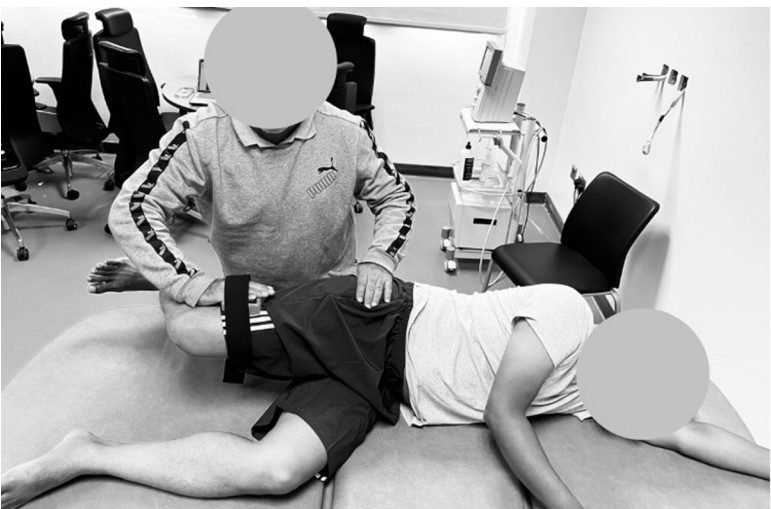

**Fig 2. Hip abductor isometric strength assessment using a digital hand-held dynamometer.**

securely grasp the sides of the examination table for stability. The examiner applied resistance at a fixed position, and the participant, in turn, exerted a maximal effort against the dynamometer while effectively countering the force applied by the examiner. Resistance was consistently administered approximately 5 cm proximal to the knee joint line, precisely at the posterior aspect of the thigh, with the specific aim of engaging hip extension (Fig 3).

The examiner articulated a standardized verbal command, directing the participant with, "Go ahead-push-push-push-push and relax," and the contraction phase maintained its duration for 5 seconds. Three consecutive measurements were taken for each muscle group, and the maximum force output (Kgf) was recorded. A trained assessor, blinded to the participant's group assignment, conducted all strength measurements.

To account for the potential impact of muscle fatigue on strength measurements, we implemented structured rest intervals, allowing participants a minimum of one minute of rest between individual test attempts and three minutes before assessing different muscle groups.

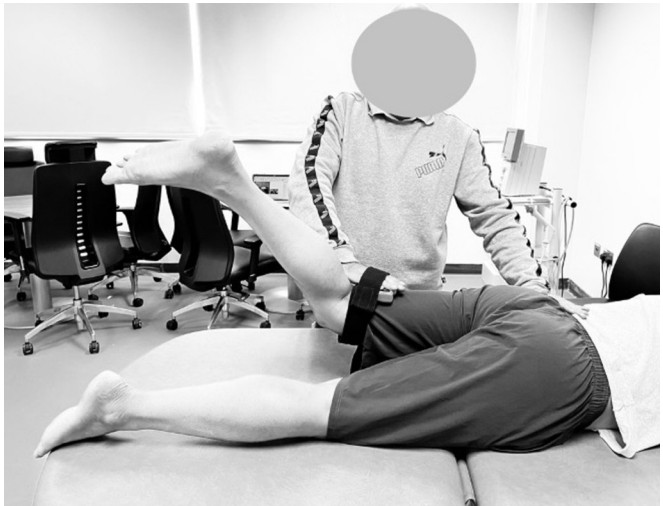

**Fig 3. Hip extensor isometric strength assessment using a digital hand-held dynamometer.**

We employed a testing sequence designed to prevent fatigue carry-over effects by alternating between non-adjacent muscle groups. Participants were also closely monitored for any signs of fatigue, with additional rest periods provided as needed. This strategy ensured that the measurements accurately reflected the participant's maximal strength capabilities.

## 2.5 Hip joint proprioception testing using digital inclinometer

In this study, we employed a digital inclinometer unit to assess hip joint proprioception (JRS). The assessments were conducted in a controlled and quiet environment, with a specific focus on individuals diagnosed with unilateral hip OA affecting one hip. For hip JRS evaluation in flexion, we positioned the digital inclinometer along the anterior and middle regions of the participant's thigh, securely fastening it with a hook-and-loop strap. Conversely, for hip JRS in abduction, the inclinometer was placed on the lateral and middle aspects of the thigh. To establish the target position for hip JRS testing, we calculated 50% of each participant's hip's full range of motion (ROM) in both flexion and abduction directions [23]. The JRS assessments were conducted exclusively in the supine position, with participants wearing blindfolds throughout the procedures.

The participants were either lying flat on the examination couch for hip flexion evaluation or positioned in a side-lying posture for hip abduction assessment. When assessing hip flexion, the primary part of the digital inclinometer was affixed to the lateral aspect of the thigh (Fig 4) and the secondary inclinometer at the waist, and for hip abduction, the primary inclinometer was securely attached to the posterior aspect of the thigh. The testing protocol involved the "active-active" reproduction technique, focusing on the hip affected by OA. Initially, participants were blindfolded, and their hip was actively moved to the target position, i.e. 60 degrees of flexion and 25 degrees of abduction. Participants were instructed to "halt" and maintain this target position for five seconds while memorizing it. Subsequently, the hip was returned to the starting position, and participants actively repositioned their hip to replicate the target angle, confirming its accuracy with a verbal affirmation of "Yes." The absolute difference, measured in degrees, between the target and reproduced angle represented the joint position error, providing an indicator of JRS precision. Each test was repeated three times, and the average of these measurements was used for subsequent analysis. The sequence of JRS testing, which

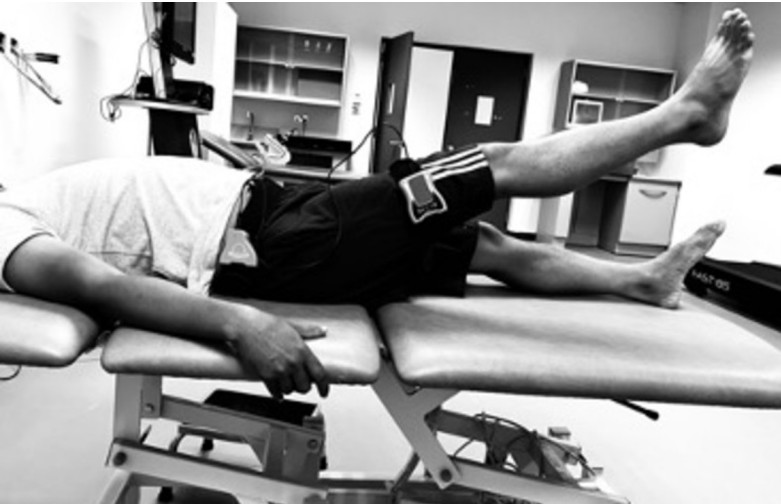

**Fig 4. Assessment of hip joint repositioning sense using a dual digital inclinometer.**

included supine positioning and assessment in both flexion and abduction directions, was randomly determined by a coin toss.

## 2.6 Functional balance assessment

Functional balance was assessed using standardized clinical tests such as the Berg Balance Scale (BBS) and Timed Up and Go (TUG) test [24, 25]. The BBS was a widely used clinical tool designed to assess and quantify an individual's static and dynamic balance abilities [26]. This scale comprised a series of 14 balance-related tasks of varying difficulty, such as standing up from a chair, reaching for objects, and maintaining balance while standing on one leg. Each task was scored on a five-point ordinal scale, with higher scores indicating better balance performance. The BBS was considered a reliable and valid measure of balance, with excellent inter-rater and intra-rater reliability. It demonstrated high sensitivity to changes in balance over time and was a valuable tool for assessing balance deficits in various populations, including those with neurological conditions, musculoskeletal disorders, and older adults. The administration of the BBS typically involved a trained clinician observing and scoring the participant's performance on each task. It provided valuable insights into an individual's balance capabilities, aiding in the development of tailored intervention plans and tracking progress in balance rehabilitation programs.

The TUG test was a commonly used clinical assessment tool for evaluating an individual's mobility and functional mobility [27]. This test involved measuring the time it took for a person to rise from a standard armchair (with armrests), walk a distance of 3 meters, turn around, walk back to the chair, and sit down [27]. The TUG test provided a simple and quick way to assess an individual's mobility and risk of falls. It was particularly valuable in identifying mobility issues in older adults and individuals with various musculoskeletal or neurological conditions. The TUG test demonstrated good test-retest reliability, making it a dependable tool for repeated assessments to monitor changes in mobility over time. Administration of the TUG test involved clear instructions to the participant to perform the task safely and as quickly as possible. The time taken to complete the task was recorded, with longer durations indicating poorer mobility and a higher risk of falling. The TUG test was widely utilized in clinical settings for its ease of use, reliability, and ability to provide valuable information about a person's functional mobility and fall risk [28].

To ensure the objectivity of our data, we employed a double-blind methodology during the collection and analysis phases. Assessors performing the muscle strength, proprioception and functional balance evaluations were blinded to the participants' group status, having no prior knowledge of whether they belonged to the Hip OA or control group. Furthermore, the statistical analysis was conducted by an independent analyst unaware of the participants' group assignments, using a numerical coding system to prevent any potential bias. These measures were crucial to maintain the integrity and impartiality of our data.

## 2.7 Data analysis

The data collected in this study underwent a thorough analysis to address the research objectives. Prior to analysis, the normality of the data distribution was assessed using appropriate statistical tests, and it was determined that the data closely followed a normal distribution. Consequently, parametric statistical methods were employed for data analysis, allowing for robust statistical inference. Descriptive statistics, such as means and standard deviations, were calculated to summarize the characteristics of the study participants, including demographic information and clinical variables. To address the primary research objectives, which included comparing hip muscle strength, hip joint proprioception, and functional balance between

individuals with unilateral hip osteoarthritis and asymptomatic individuals, independent t-tests were conducted. This parametric test allowed for the evaluation of mean differences between the two groups. Furthermore, correlation analyses, such as Pearson correlation coefficients, were employed to investigate potential relationships between hip muscle strength, hip joint proprioception, and functional balance in individuals with unilateral hip osteoarthritis. This analysis aimed to identify any significant associations between these variables. To explore the potential mediating roles of hip muscle strength and hip joint proprioception in the relationship between unilateral hip osteoarthritis and functional balance, mediation analysis was performed using multiple linear regression analysis. All statistical analyses were carried out using statistical software packages (SPSS, version 20.0), and the level of significance was set at $p < 0.05$.

## 3. Results

The demographic characteristics of the Unilateral Hip OA and Asymptomatic populations, comprising 62 individuals each, were compared in Table 1. The Unilateral Hip OA group had a slightly higher mean age (68.1 years ± 4.5) compared to the asymptomatic group (65.7 years ± 3.9), with both groups having a similar gender distribution. The mean BMI was 26.4 kg/m$^2$ ± 3.3 for Unilateral Hip OA and 25.8 kg/m$^2$ ± 3.1 for asymptomatic individuals. The Unilateral Hip OA group reported a mean VAS pain score of 54.76 mm ± 11.18, while the asymptomatic group had no recorded VAS pain scores. Similarly, the HOOS scores were available only for the Unilateral Hip OA group, with a mean of 60.8 ± 9.7. These results provide a snapshot of the demographic and pain-related characteristics of the two populations under study.

Table 2 presents a comparison of muscle strength, hip JRS, and functional balance between the Hip OA and Control groups. In the Hip OA group, mean values for hip flexor, hip extensor, and hip abductor strength were 11.68 ± 2.98 Kgf, 10.24 ± 3.22 Kgf, and 09.34 ± 3.98 Kgf, respectively, while in the Control group, these values were 15.47 ± 3.05 Kgf, 13.87 ± 4.23 Kgf, and 12.45 ± 3.45 Kgf. Statistically significant differences were observed in all three strength measures, with p-values of 0.003, 0.002, and 0.001, respectively, indicating that the Control group had higher strength levels. Hip JRS in flexion and abduction was also significantly different between the two groups, with p-values less than 0.001. Functional balance, as assessed by the BBS and BBS test, showed substantial differences, with p-values less than 0.001. The effect sizes, calculated as Cohen's d, indicate moderate to large differences in muscle strength and functional balance measures between the groups.

**Table 1. Demographic characteristics of the unilateral hip OA and asymptomatic populations.**

| Variable | Unilateral Hip OA (n = 62) | Asymptomatic (n = 62) |
|---|---|---|
| | (Mean ± SD) | |
| Age (years) | 68.1 ± 4.5 | 65.7 ± 3.9 |
| Gender (Male/Female) | 32 M / 30 F | 34 M / 28 F |
| BMI (Kg/m2)[a] | 26.4 ± 3.3 | 25.8 ± 3.1 |
| VAS [b] pain score (0–100 mm) | 54.76 ± 11.18 | - |
| HOOS [c] (0–100 score) | 60.8 ± 9.7 | - |

[a] BMI: Body Mass Index

[b] VAS: Visual Analogue Scale

[c] HOOS: Hip Disability and Osteoarthritis Outcome Score

**Table 2. Comparison of muscle strength, hip joint reposition sense, and functional balance between Hip OA and control groups.**

| | Variables | Hip OA (Mean ± SD) | Control (Mean ± SD) | p -Value | Mean Difference (Mean ± SD) | 95% CI of Mean Difference[c] | Difference—Effect Size |
|---|---|---|---|---|---|---|---|
| Muscle strength | Hip Flexors (Kgf) | 11.68 ± 2.98 | 15.47 ± 3.05 | 0.003 | 4.25 ± 0.12 | 7.01, -20.01 | 0.76 |
| | Hip Extensors (Kgf) | 10.24 ± 3.22 | 13.87 ± 4.23 | 0.002 | 3.11 ± 0.11 | 6.03, -17.09 | 0.81 |
| | Hip Abductors (Kgf) | 09.34 ± 3.98 | 12.45 ± 3.45 | 0.001 | 3.98 ± 0.09 | 4.09, -16.45 | 0.89 |
| Proprioception | Hip JRS in flexion (°) | 4.96 ± 1.09 | 2.45 ± 0.23 | <0.001 | 2.97 ± 0.98 | -1.34, 8.87 | 0.76 |
| | Hip JRS in Abduction (°) | 5.67 ± 1.88 | 2.56 ± 0.98 | <0.001 | 2.89 ± 1.08 | -1.45, 9.07 | 0.84 |
| Functional Balance | BBS Score | 39.43 ± 6.34 | 52.3 ± 3.56 | <0.001 | 11.34 ± 3.01 | 31.67, 46.76 | 0.67 |
| | TUG Score | 10.98 ± 3.34 | 6.98 ± 2.45 | | 3.34 ± 0.98 | 06.09, 14.98 | 0.58 |

[a]Kgf: kilogram-force

[b]JRS: joint reposition sense

[c]Effect size is calculated as Cohen's d.

Table 3 presents the correlation between hip muscle strength, hip JRS, and functional balance in Hip OA individuals. The results show that there is a mild to moderate positive correlation between muscle strength (hip flexors, hip extensors, and hip abductors) and functional balance, as indicated by the BBS score. The correlation coefficients (r) range from 0.38 to 0.42, and the p-values are all significant (p < 0.003), suggesting that stronger hip muscles are associated with better functional balance in Hip OA individuals. Conversely, there is a mild to moderate negative correlation between muscle strength and the BBS test score, with correlation coefficients ranging from -0.36 to -0.41 and p-values less than 0.003. This indicates that weaker hip muscles are associated with slower performance on the TUG test. Additionally, there is a mild to moderate negative correlation between hip JRS in flexion and functional balance (r = -0.46, p < 0.001) and a mild to moderate positive correlation between hip JRS in abduction and functional balance (r = 0.46, p < 0.001), suggesting that impaired hip JRS in flexion is associated with worse functional balance, while improved hip JRS in abduction is associated with better functional balance in Hip OA individuals.

## 4. Discussion

The study aimed to assess the degree of impairment in hip muscle strength, hip joint proprioception, and functional balance in individuals with unilateral Hip OA, comparing these

**Table 3. Correlation between hip muscle strength, hip joint reposition sense, and functional balance in Hip OA individuals.**

| Variables | BBS score | | TUG-Test score | |
|---|---|---|---|---|
| | r | p-value | r | p-value |
| Hip Flexors (Kgf[a]) | 0.38 | 0.002 | -0.39 | 0.003 |
| Hip Extensors (Kgf) | 0.42 | <0.001 | -0.41 | 0.001 |
| Hip Abductors (Kgf) | 0.39 | 0.001 | -0.36 | 0.003 |
| Hip JRS[b] in flexion (°) | -0.46 | <0.001 | 0.45 | <0.001 |
| Hip JRS in Abduction (°) | -0.49 | <0.001 | 0.46 | <0.001 |

[a] Kgf: kilogram-force

[b] JRS: joint reposition sense

parameters to asymptomatic individuals. The objectives were twofold: firstly, to compare these key musculoskeletal health indicators between the two groups, and secondly, to investigate the interrelations among them within the Hip OA cohort. The results demonstrated significant deficits in hip muscle strength and proprioceptive accuracy in the OA group compared to the controls, highlighting the extent of musculoskeletal impairment. Furthermore, the study identified meaningful correlations between muscle strength and functional balance, suggesting that the strength of hip muscles is a predictor of balance performance in individuals with unilateral Hip OA. These findings provide a foundation for the discussion on the implications of hip muscle strength and proprioception on functional balance and the overall management of Hip OA.

The study data presented showcases significant distinctions in muscle strength, hip JRS, and functional balance between the Hip OA and control groups. The observed differences in muscle strength complement the assertions of previous studies, which have identified muscle weakness as a prevalent feature in individuals with Hip OA [29, 30]. The significant variations in hip flexor, extensor, and abductor strength among our participants mirror the recognized pattern of muscular impairment associated with this condition [29, 30]. Muscle strength, particularly in the hip flexors, extensors, and abductors, is frequently compromised in individuals with Hip OA, as reported by numerous studies [29, 30]. These studies have elucidated that muscle weakness in OA patients is often a consequence of pain, muscle atrophy, and altered neuromuscular function [31, 32]. In line with existing literature, the Hip OA group's compromised hip JRS reflects the degeneration of proprioceptive function commonly reported in osteoarthritis patients [33, 34]. This sensory decline is known to adversely affect joint stability and may exacerbate the risk of injury due to falls, as proprioception plays a crucial role in maintaining balance and coordinating movements [4, 35]. Furthermore, the decline in hip JRS observed in the Hip OA group resonates with the findings of Wingert et al., [36] who noted that proprioceptive deficits were prominent among OA sufferers due to the deterioration of sensory receptors in joint tissues [36]. These proprioceptive deficits are pivotal as they contribute to the challenges in balance and joint stability, corroborating with findings from Benjaminse et al. [37], who emphasized the proprioceptive loss in OA patients as a key factor influencing postural control [37].

Functional balance, as measured by standard clinical tests, also showed marked differences between the two groups. These findings are not unexpected, given that previous research has consistently shown a correlation between lower extremity joint pathologies and impaired balance [38, 39]. The significance of these differences is not merely statistical but extends to the real-world impact on individuals' functional abilities and their potential risk of falls [39]. The results regarding functional balance, as assessed by the BBS and the BBS test, are in line with those of Hicks et al. [39], who demonstrated that individuals with lower extremity OA had poorer balance and increased fall risk [39]. The clinically significant differences in balance performance between our Hip OA group and the control group underscore the functional implications of OA on daily activities and risk of falls, a concern also reflected in the works of Knox et al. [40].

The effect sizes for muscle strength and functional balance suggest that these impairments are not only statistically significant but also clinically meaningful. These results underscore the necessity for therapeutic strategies that prioritize muscle strengthening, balance training, and proprioceptive enhancement for individuals with Hip OA [41, 42]. Such interventions are crucial for improving the quality of life and functional independence in this population [42]. The moderate to large effect sizes reported in our analysis for both muscle strength and functional balance measures are indicative of the real-world impact of these deficits [43]. This notion is supported by Loureiro et al. [31], who argued that even moderate changes in muscle function could have significant functional implications for individuals with Hip OA [31]. Their research

posits that therapeutic interventions should be targeted towards improving these specific deficits to enhance overall functional outcomes [44].

The observed correlations between muscle strength in the hip musculature and functional balance metrics offer significant insights into the interdependencies of musculoskeletal functions in individuals with Hip OA [44]. The observed correlations between muscle strength and functional balance in individuals with Hip OA can be largely explained by the integral role of the hip musculature in maintaining postural stability and controlling body movements [44–46]. Stronger hip flexors, extensors, and abductors contribute to a stable pelvis and a secure base of support, which are essential for good balance, as reflected in higher BBS scores [47]. This relationship is further evidenced by the significant positive correlation between muscle strength and BBS scores [47]. The hip muscles are particularly important for performing functional movements efficiently, such as those required in the BBS test, explaining the negative correlation observed between muscle strength and TUG scores [48]. The mild to moderate positive correlations between hip muscle strength (flexors, extensors, and abductors) and the BBS scores corroborate with previous findings suggesting that muscle strength is a key determinant of balance [48–50]. These studies have consistently shown that hip muscles play a pivotal role in stabilizing the pelvis and maintaining postural control, which are critical for balance [50]. The significant correlation coefficients ranging from 0.38 to 0.42 align with the literature, reinforcing the notion that hip muscle strength is fundamentally linked to balance performance [51, 52]. The strength of the hip musculature, particularly the abductors, has been previously associated with balance due to their role in lateral stability [53]. The statistical significance of these correlations ($p < 0.003$) in our study further substantiates the importance of targeting these muscle groups in therapeutic interventions for Hip OA. Conversely, the mild to moderate negative correlations between muscle strength and the BBS test scores suggest that weaker hip muscles are associated with increased times on the TUG test, reflecting slower functional mobility [54, 55]. This is in line with studies such as those by Murao et al. [56], who found that hip strength, particularly in the extensors and abductors, was predictive of performance in functional tasks like the TUG test [56]. The correlation coefficients ranging from -0.36 to -0.41, with significant p-values ($p < 0.003$), emphasize the clinical relevance of hip muscle strength in tasks that require both static and dynamic balance.

The negative correlation between hip JRS in flexion and functional balance suggests that proprioceptive deficits can adversely affect balance. In individuals with Hip OA, the degradation of these receptors can impair proprioceptive acuity, particularly in hip flexion, which is vital for balance and stability [11]. The negative correlation between hip JRS in flexion and functional balance underscores this point, suggesting that proprioceptive deficits can lead to poorer balance outcomes [57]. Conversely, better proprioceptive function in hip abduction is associated with improved balance, as indicated by the positive correlation with functional balance measures [58]. The presence of OA itself brings about several changes that can contribute to these findings [58]. The pathological changes in the joint structure, including cartilage wear and bone alterations, can lead to altered joint mechanics and a decrease in proprioceptive sensitivity [3]. Furthermore, the pain and inflammation characteristic of OA can cause muscle inhibition and atrophy, reducing muscle strength and further compromising joint function and stability [59, 60]. This finding is consistent with research highlighting the role of proprioception in joint stability and balance [10, 61]. The correlation coefficient of -0.46 ($p < 0.001$) indicates a moderate relationship, where worse hip JRS in flexion, which may result from the sensory deficits associated with OA, corresponds to poorer balance outcomes. Conversely, the positive correlation between hip JRS in abduction and functional balance, with a similar coefficient and significance level, underscores the positive role of abduction proprioception in maintaining balance [61]. This finding may reflect the importance of abduction movements in

lateral weight shifting, which is essential for balance during gait and other functional activities [49, 62]. These findings are consistent with a breadth of previous research. Studies have demonstrated a direct link between lower limb muscle strength and balance, with a focus on the importance of the hip muscles [31, 49]. Additionally, the adverse effects of OA on proprioception have been well documented, showing a clear connection with declines in postural control and gait patterns [10, 49]. Moreover, the association between muscle weakness and functional performance in tasks requiring balance and mobility has been reported [10, 49].

While our study provided valuable insights into the strength deficits of the hip flexors, extensors, and abductors in individuals with hip osteoarthritis, we acknowledge the exclusion of internal and external rotation strength measurements as a limitation. Internal and external rotation is important for a range of motions and activities, including gait dynamics and transitions between movements [63]. However, given their less direct impact on the primary functional limitations of hip OA and the increased methodological complexity required for accurate assessment, they were not included in our primary analysis [63]. Future studies could benefit from incorporating these measurements to provide a more comprehensive profile of muscular impairment in hip osteoarthritis. Additionally, investigating the rotational strength may offer further insights into the pathophysiology of OA and the full spectrum of functional limitations experienced by patients, potentially guiding more targeted therapeutic interventions [63].

## 4.1 Clinical significance

This study elucidates the clinical relevance of muscular strength, proprioception, and functional balance in Hip OA management, emphasizing the need for comprehensive therapeutic approaches. Highlighting the impact of muscle weakness on balance and the risks of falls, the research corroborates previous findings that pain, atrophy, and neuromuscular dysfunction contribute to strength deficits, necessitating focused clinical intervention [10, 49]. The study also demonstrates significant proprioceptive declines in Hip OA, advocating for proprioceptive training to improve balance outcomes. Differences in functional balance between Hip OA patients and controls, and the relationship between muscle strength and balance performance, underscore the value of strength and balance training in improving mobility and quality of life. The moderate to large effect sizes for these measures support the implementation of targeted therapeutic interventions to enhance functional outcomes, advocating for early intervention to maintain muscle function and proprioception, ultimately aiming to decelerate functional deterioration and elevate patient care.

## 4.2 Areas of future research

Future research should focus on longitudinal studies to delineate the progression of muscular and proprioceptive decline in Hip OA and evaluate the long-term efficacy of targeted interventions such as strength, proprioceptive, and balance training. It would be valuable to explore the molecular and biomechanical pathways underlying muscle weakness and proprioceptive loss, to better understand the pathophysiology of OA. Additionally, there is a need for randomized controlled trials to establish the optimal types and intensities of rehabilitation exercises that maximize functional outcomes. Investigating the potential benefits of integrating technology, like wearables and virtual reality, in rehabilitation programs could also offer innovative approaches to managing Hip OA. Finally, studies assessing the psychosocial impacts of muscle strength and balance impairments on individuals with Hip OA could provide insights into holistic care approaches that address both the physical and emotional well-being of patients.

### 4.3 Limitations

The present study, while comprehensive in its scope, does have several limitations that must be acknowledged. Firstly, the reliance on specific clinical measures to assess functional balance and proprioception, which may not capture the full spectrum of balance-related impairments experienced by individuals with Hip OA. Additionally, the study did not account for the potential influence of medication, pain severity, or the duration of OA, which can all impact muscle function and balance. Also, the lack of detailed data on participants' physical activity levels and other comorbidities prevents a more nuanced understanding of the complex interactions that may influence the observed outcomes. Future studies should aim to address these limitations by employing longitudinal designs, larger and more diverse populations, and a broader range of diagnostic tools to elucidate the dynamics of musculoskeletal health in Hip OA.

## 5. Conclusions

The study's conclusion emphasizes that individuals with unilateral Hip OA display notable impairments in muscle strength, proprioception, and functional balance compared to asymptomatic individuals. Furthermore, it was found that hip muscle strength is significantly correlated with functional balance. This underscores the importance of focused muscle strengthening and proprioceptive training in addressing balance problems in patients with Hip OA. The insights gained from this research enhance our comprehension of the musculoskeletal impacts of Hip OA and will guide the development of clinical interventions to improve functional mobility and life quality for patients suffering from this condition.

## Supporting information

**S1 Raw data.**
(XLSX)

## Author Contributions

**Conceptualization:** Batool Abdulelah Alkhamis, Ravi Shankar Reddy, Khalid A. Alahmari, Mastour Saeed Alshahrani, Ghada Mohammed Koura, Olfat Ibrahim Ali, Debjani Mukherjee, Basant Hamdy Elrefaey.

**Data curation:** Batool Abdulelah Alkhamis, Ravi Shankar Reddy, Khalid A. Alahmari, Mastour Saeed Alshahrani, Ghada Mohammed Koura, Olfat Ibrahim Ali, Debjani Mukherjee, Basant Hamdy Elrefaey.

**Formal analysis:** Batool Abdulelah Alkhamis, Ravi Shankar Reddy, Ghada Mohammed Koura.

**Funding acquisition:** Batool Abdulelah Alkhamis.

**Investigation:** Batool Abdulelah Alkhamis, Ravi Shankar Reddy.

**Methodology:** Batool Abdulelah Alkhamis, Ravi Shankar Reddy, Khalid A. Alahmari, Mastour Saeed Alshahrani, Ghada Mohammed Koura, Debjani Mukherjee, Basant Hamdy Elrefaey.

**Project administration:** Batool Abdulelah Alkhamis.

**Resources:** Batool Abdulelah Alkhamis.

**Software:** Batool Abdulelah Alkhamis.

**Writing – original draft:** Batool Abdulelah Alkhamis, Ravi Shankar Reddy, Khalid A. Alahmari, Mastour Saeed Alshahrani, Basant Hamdy Elrefaey.

**Writing – review & editing:** Batool Abdulelah Alkhamis, Ravi Shankar Reddy, Khalid A. Alahmari, Mastour Saeed Alshahrani, Ghada Mohammed Koura, Olfat Ibrahim Ali, Debjani Mukherjee, Basant Hamdy Elrefaey.

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
