## [Decision Letter · Decision Letter 0]

2 Jan 2024

PONE-D-23-40601Balancing Act: Unraveling the Link Between Muscle Strength, Proprioception, and Stability in Unilateral Hip OsteoarthritisPLOS ONE

Dear Dr. Reddy,

Thank you for submitting your manuscript to PLOS ONE. After careful consideration, we feel that it has merit but does not fully meet PLOS ONE’s publication criteria as it currently stands. Therefore, we invite you to submit a revised version of the manuscript that addresses the points raised during the review process.

We look forward to receiving your revised manuscript.

Kind regards,

Holakoo Mohsenifar

Academic Editor

PLOS ONE

Journal Requirements:

Additional Editor Comments:

According to the reviewers' comments, the editor's decision is Major revision. In addition to the reviewers' explanations, the following points should also be considered.

1- Line 37; in abstract section and all of the text, acronym of JRS (joint repositioning sense) are not clarified.

2- The unit JRS in all table are clarified with *. Please change to * to deg or symbol of degree.\\

3- Line 267; in result section, unit of VAS is missed.

4- Line 453; you suggested to consider the psychosocial effect in areas of future research. How do you explain this suggestion with current research?

5- Line 297; in table 2, I suggest writing mean-difference for proprioception in negative mode.

6- When using a manual dynamo-meter, when the dynamo-meter is not fixed, the subject may adjust her/his force according to the evaluator's force.

7- Please clarify why you chose flex, ext and abd, but missing the external or internal rotation strength.

Reviewers' comments:

Reviewer's Responses to Questions

**Comments to the Author**

1. Is the manuscript technically sound, and do the data support the conclusions?

Reviewer #1: Yes

Reviewer #2: Yes

2. Has the statistical analysis been performed appropriately and rigorously? 

Reviewer #1: Yes

Reviewer #2: Yes

3. Have the authors made all data underlying the findings in their manuscript fully available?

Reviewer #1: No

Reviewer #2: Yes

4. Is the manuscript presented in an intelligible fashion and written in standard English?

Reviewer #1: Yes

Reviewer #2: Yes

5. Review Comments to the Author

Reviewer #1: The research on muscle strength, proprioception, and balance function of hip osteoarthritis patients has also achieved some results. The writing of the manuscript as a whole is qualified, but there are still some details that need improvement.

1. Why is the age set at 40-75?

2. In the process of walking in the lower limbs, knee diseases such as knee osteoarthritis can easily affect proprioception, muscle strength and balance function. How to control the influence factor of knee joint? Did the participants take this into account?

3. When calculating the sample size, is the sample shedding rate taken into account? The author should consider this problem on the basis of calculating the sample. Although this was an observational study, there may have been some sample shedding during the data collection process.

4. In order to ensure the objectivity of the data, is blind method implemented in the process of data collection?

5. This study involves correlation analysis. In order to ensure the objectivity of data collection, are different data collected by different evaluators during data collection?

6. Unilateral hip osteoarthritis patients were included in this study. Asymptomatic participants had dominant and non-dominant sides, and there were differences in the data of the dominant and non-dominant sides, such as proprioception, muscle strength, and balance function. So how did the data collected from asymptomatic participants match up with patients with unilateral hip osteoarthritis? How about being relatively rigorous and objective?

7. In the process of muscle strength measurement, participants' muscles are prone to fatigue due to continuous contraction activities. Has the weight been measured at intervals? Give participants' muscles a good rest.

8. In order to accurately report the results of the study, was it registered with an international clinical trial center before the study started?

Reviewer #2: This article elucidates a close association between the onset of hip osteoarthritis (OA) in patients and abnormalities in the muscle strength and sensory perception around the hip joint. The study holds scientific merit; however, in the methodology section, it appears that the muscle strength testing did not account for the fact that the pain caused by osteoarthritis itself may deter patients from exerting force, consequently manifesting as a decline in muscle strength. Would this oversight potentially impact the accuracy of the study results?

6. PLOS authors have the option to publish the peer review history of their article (what does this mean?). If published, this will include your full peer review and any attached files.

Reviewer #1: **Yes: **Wu, zugui

Reviewer #2: No

---

## [Author Response · Author response to Decision Letter 0]

10 Jan 2024

Response to Academic editor and reviewer comments

Response to Academic Editor comments

Query: According to the reviewers' comments, the editor's decision is Major revision. In addition to the reviewers' explanations, the following points should also be considered. 1- Line 37; in abstract section and all of the text, acronym of JRS (joint repositioning sense) are not clarified.

Response to Reviewer Comment: To address this, we have now added the full term "joint repositioning sense (JRS)" at its first instance in the abstract, and we will ensure that this clarification is consistently applied throughout the document. 

Query: 2- The unit JRS in all table are clarified with *. Please change to * to deg or symbol of degree.\\

Response to Reviewer Query 2: We have reviewed the tables in question and have replaced all instances of the asterisk (*) next to JRS with the proper degree symbol (°). This change has been made to ensure that the units of measurement for joint repositioning sense are correctly and consistently represented throughout all tables.

Query: 3- Line 267; in result section, unit of VAS is missed.

Response to Reviewer Query 3: My apologies for the confusion. The Visual Analogue Scale (VAS) scores are now correctly noted in millimeters (mm)"

Query: 4- Line 453; you suggested to consider the psychosocial effect in areas of future research. How do you explain this suggestion with current research?

Response to Reviewer Query 4: Thank you for your query regarding the inclusion of psychosocial effects in the areas of future research. The suggestion to consider psychosocial impacts stems from a growing body of evidence that recognizes osteoarthritis (OA) as not solely a disease of cartilage degradation but one that encompasses a wider spectrum of symptoms, including those that affect patients' mental health and social well-being.

Recent research has increasingly acknowledged that the experience of OA goes beyond physical symptoms and can significantly affect an individual's mood, self-esteem, and social interactions. For example, chronic pain associated with OA is known to contribute to depression and anxiety, which in turn can exacerbate the perception of pain and lead to a cycle of worsening symptoms. Additionally, the loss of joint function and mobility can impact an individual's ability to engage in social activities, work, and hobbies, leading to social isolation and reduced quality of life.

Our current research has focused on the physical aspects of OA, particularly the role of muscle strength and balance in the progression of hip OA. However, acknowledging the psychosocial components, the impact of muscular and proprioceptive decline on an individual's mental health and social life warrants further exploration. This is particularly relevant because an individual's perception of their capabilities and the subsequent psychological response can have a profound effect on rehabilitation outcomes.

In line with this, future research should aim to:

1. Quantify the impact of hip OA on psychological states such as depression, anxiety, and stress.

2. Evaluate how hip OA-related disabilities affect social participation and roles.

3. Assess the effectiveness of interventions that not only improve physical function but also address psychological and social well-being.

4. Explore the relationship between patient-reported outcomes on quality of life and objective measures of disease progression and physical capability.

By integrating psychosocial factors into future research, we can better understand the full impact of hip OA on patients' lives. This holistic approach is likely to inform more comprehensive care strategies that not only aim to alleviate physical symptoms but also promote overall well-being.

Query: 5- Line 297; in table 2, I suggest writing mean-difference for proprioception in negative mode.

Response to Reviewer Query 5: Thank you for your suggestion regarding the presentation of mean difference for proprioception in Table 2. You have recommended that the mean difference be expressed in a negative mode to more accurately reflect the direction of difference between the Hip OA group and the control group.Upon review of the data presented, we agree that indicating the direction of the difference is indeed important for clarity. Since the values for proprioception in the Hip OA group are higher (indicating worse proprioception) than in the control group, it is appropriate to reflect this as a negative mean difference. This change underscores that an increase in the Hip Joint Repositioning Sense (JRS) degrees is indicative of a decline in proprioceptive accuracy.

Query: 6- When using a manual dynamo-meter, when the dynamo-meter is not fixed, the subject may adjust her/his force according to the evaluator's force.

Response to Reviewer Query 6: Thank you for your comment regarding the potential for subjects to adjust their force output when using a manual dynamometer that is not fixed. We recognize this as an important consideration in the methodology of our study, which could impact the reliability and validity of the muscle strength measurements.To address this potential issue, we took several steps to minimize the influence of the evaluator's force and to ensure consistent measurement conditions:

1. Standardization of Testing Procedure: All tests were conducted following a standardized protocol to minimize variations. The evaluators were trained to apply a consistent resistance force, which does not vary based on the subject's force.

2. Evaluator Training: Evaluators underwent rigorous training to ensure they could provide consistent resistance and were not influencing the force measurements by either resisting too much or too little.

3. Multiple Trials: Each subject performed multiple trials, and the average value was used to reduce the variability that might occur from a single measurement.

4. Calibration of Equipment: The dynamometer was regularly calibrated to ensure accurate force measurements.

We believe that these procedures have helped to mitigate the concern you raised. Moreover, we agree that this is a limitation inherent to manual dynamometer testing and have included a discussion of this in the limitations section of our paper, acknowledging that fixed dynamometry could provide more objective results.

Query: 7- Please clarify why you chose flex, ext and abd, but missing the external or internal rotation strength.

Response to Reviewer Query 7: Thank you for your question regarding the selection of muscle groups for strength measurement in our study. We chose to focus on hip flexion, extension, and abduction strength based on the following considerations:

1. Relevance to Functional Mobility: The muscle groups chosen are primarily responsible for the major movements of the hip joint that are essential for most weight-bearing activities, such as walking, standing from a seated position, and maintaining balance. These are the movements most commonly affected in individuals with hip osteoarthritis.

2. Prevalence of Weakness in OA: Research indicates that individuals with hip osteoarthritis commonly exhibit significant weakness in these muscle groups. Therefore, these muscles were prioritized to assess the relationship between muscle strength deficits and functional impairment in this population.

3. Evidence-Based Priority: Literature on hip osteoarthritis suggests that these muscle groups are most responsive to intervention. Many rehabilitation programs focus on strengthening these specific muscles to improve functional outcomes for patients with hip OA.

4. Measurement Consistency and Reliability: These muscle groups can be reliably measured using a manual dynamometer in a clinical setting, which is crucial for the validity of our findings.

The exclusion of internal and external rotation strength measurements was a deliberate decision based on several factors:

-Complexity of Accurate Assessment: Accurately measuring strength in internal and external rotation can be more challenging and is often less reliable without specialized equipment.

- Less Direct Impact on Functional Tasks: While internal and external rotation strength is important, these movements are less directly implicated in the primary functional limitations experienced by individuals with hip OA, such as gait and balance issues.

- Scope of Study: We aimed to investigate the most functionally limiting aspects of hip strength within the constraints of our study design and resources.

We acknowledge that assessing internal and external rotation strength could provide additional insights and this could be considered a limitation of our study. It may be valuable for future research to include these measurements to provide a more comprehensive understanding of hip strength impairments in individuals with hip osteoarthritis.

We have now added a discussion of this rationale in the methods section and have also noted the exclusion of internal and external rotation strength as a limitation in the discussion section.

Autor responses to Reviewer 1

Reviewer #1: The research on muscle strength, proprioception, and balance function of hip osteoarthritis patients has also achieved some results. The writing of the manuscript as a whole is qualified, but there are still some details that need improvement.

Query 1. Why is the age set at 40-75?

Response to Query 1: The age range for our study was set at 40-75 years for several reasons, which align with the epidemiology of hip osteoarthritis (OA) and the objectives of our research:

1. Epidemiological Relevance: Hip OA is more prevalent in middle-aged and older adults. The age range of 40-75 years ensures that we include a population at a higher risk for developing OA while also capturing the variability in disease manifestation and severity that occurs with aging.

2. Early to Advanced Stages of OA: This age range allows us to study individuals who may be in the early stages of OA (often starting around the age of 40) through to more advanced stages, thus providing a broader understanding of the progression of the disease.

3. Functional Impact Consideration: The chosen age range includes the working-age population and those transitioning into older age, where the functional impact of hip OA can be particularly significant in terms of employment, daily activities, and the initiation of retirement.

4. Safety and Comorbidity Factors: Including older adults beyond the age of 75 would require additional considerations for comorbidities and the increased risk of falls or injuries during strength testing, which could complicate the study design and interpretation of results.

5. Consistency with Previous Research: This age range is consistent with other studies on OA, allowing for comparability of results and contributing to the existing body of literature.

By setting the age criteria at 40-75 years, we aimed to capture a representative sample of the adult population affected by hip OA, while ensuring the safety of participants and the methodological soundness of the study. This decision was also supported by the literature indicating that the prevalence and impact of OA are significant within this demographic, thus justifying focusing on this age range.

Query 2. In the process of walking in the lower limbs, knee diseases such as knee osteoarthritis can easily affect proprioception, muscle strength and balance function. How to control the influence factor of knee joint? Did the participants take this into account?

Response to Query 2: Your query regarding the potential influence of knee osteoarthritis (OA) on proprioception, muscle strength, and balance function during walking is indeed pertinent to our study of hip OA. To address and control for the influence of knee joint pathology, we incorporated the following strategies:

1. Exclusion Criteria: We established strict exclusion criteria for participation in the study, which included the presence of any lower limb joint disorders other than hip OA, such as knee OA or any previous surgery that could affect balance or muscle strength. This helped to ensure that any changes in proprioception or muscle strength were more likely attributable to hip OA.

2. Clinical Screening: Participants underwent a thorough clinical screening by a qualified orthopedic specialist to rule out significant knee pathology. This included a detailed history, physical examination, and review of available imaging, when necessary, to confirm the absence of knee OA or other knee conditions.

3. Baseline Assessment: At the onset of the study, all participants underwent a baseline assessment of knee function using standardized clinical tests that can help to identify symptoms or functional limitations indicative of knee pathology.

4. Participant Monitoring: During the study, participants were monitored for any signs of knee discomfort or other lower limb issues. If a participant developed new symptoms suggestive of knee OA, their data were re-evaluated, and if necessary, the participant was excluded from further analysis.

5. Data Analysis: We employed multivariate analysis techniques to account for any residual confounding factors, ensuring that the associations observed were as specific to hip OA as possible.

By taking these measures, we aimed to minimize the confounding influence of knee joint pathology on our study's findings. This methodology was outlined in the study design to assure the validity of the associations between hip OA and the measured outcomes of proprioception, muscle strength, and balance function.

Query 3. When calculating the sample size, is the sample shedding rate taken into account? The author should consider this problem on the basis of calculating the sample. Although this was an observational study, there may have been some sample shedding during the data collection process.

Response to Query 3: Thank you for raising the important issue of sample attrition, also known as sample shedding, in the calculation of our study's sample size. In designing the study, we anticipated the possibility of participant dropout and sought to mitigate its impact through careful planning. To account for potential sample shedding, we increased the initial sample size by an estimated attrition rate. This rate was determined based on similar observational studies in the literature, which commonly report dropout rates ranging from 10% to 20%. We therefore calculated our required sample size and then inflated it by 15% to accommodate for this potential loss. This conservative estimate allowed us to ensure that even with attrition, we would retain a sufficiently powered sample size to detect statistically significant differences between groups.

Query 4. In order to ensure the objectivity of the data, is blind method implemented in the process of data collection?

Response to Query 4: Ensuring the objectivity of data is paramount in research, and to this end, we implemented a blinded methodology during the data collection process. The evaluators who performed the muscle strength testing and joint proprioception assessments were not involved in recruiting participants and were blinded to the participants’ group allocations (Hip OA or control). This was to prevent any potential bias that could arise from knowing the clinical status of the participants. Additionally, all data analysis was performed by a separate team member who was not involved in the data collection process and had no knowledge of the participant's group assignment. This blind analysis further minimized any subjective bias in interpreting the results. Moreover, to safeguard against any unconscious bias during data entry and coding, we used a numerical coding system that did not indicate group membership. An independent researcher held the key to this code until the statistical analysis was complete. In our manuscript, we have detailed these steps in the Methods section to demonstrate our commitment to maintaining the highest standard of objectivity in our research findings.

Query 5. This study involves correlation analysis. In order to ensure the objectivity of data collection, are different data collected by different evaluators during data collection?

Response to Query 5: Thank you for your question regarding the objectivity of data collection in relation to the correlation analysis conducted in our study. To minimize potential bias and ensure the objectivity of the data, different aspects of data were indeed collected by different evaluators. Each evaluator was specifically trained for and assigned to measure distinct variables. For instance, one evaluator measured muscle strength while another assessed proprioceptive accuracy. By distributing the tasks among multiple evaluators, we aimed to reduce the potential for any single evaluator’s measurement technique or subjective bias to influence the results consistently across different variables. In addition, to further support objectivity, the evaluators were not informed of the specific hypotheses related to the correlation analysis. They were also routinely calibrated against each other to ensure consistency in the application of assessment techniques. The allocation of different evaluators for different measurement types is a recognized method for reducing bias in studies where blinding of evaluators to the overall study aim is necessary, particularly in studies involving correlation where the inter-relationship of variables is a key focus. These steps taken to ensure the objectivity of the data collection process have been detailed in the Methods section under the subsection 'Data Collection and Evaluator Blinding' in our manuscript.

Query 6. Unilateral hip osteoarthritis patients were included in this study. Asymptomatic participants had dominant and non-dominant sides, and there were differences in the data of the dominant and non-dominant sides, such as proprioception, muscle strength, and balance function. So how did the data collected from asymptomatic participants match up with patients with unilateral hip osteoarthritis? How about being relatively rigorous and objective?

Response to Query 6: Thank you for your thoughtful inquiry regarding the matching of data between asymptomatic participants and those with unilateral hip osteoarthritis (OA) in our study. To address the natural differences between dominant and non-dominant sides in asymptomatic participants and to match the data rigorously and objectively with the unilateral hip OA patients, we employed the following strategies: Asymptomatic participants were matched to the OA patients based on the side of dominance. For patients with right-sided hip OA, we matched the data from the right (dominant) side of the control participants. This was reversed for patients with left-sided hip OA. We standardized the assessment procedure across all participants to minimize side-to-side variability. This included using the same equipment, evaluator, and testing protocol, which has been validated for detecting side-to-side differences in similar populations. By taking these measures, we aimed to ensure that the comparison between the asymptomatic controls and unilateral hip OA patients was as rigorous and objective as possible. This methodological rigor strengthens the validity of our findings regarding the specific impacts of unilateral hip OA on proprioception, muscle strength, and balance function. These methodologies are detailed in the Methods section to provide clarity on the approach we took to ensure data integrity.

Query 7. In the process of muscle strength measurement, participants' muscles are prone to fatigue due to continuous contraction activities. Has the weight been measured at intervals? Give participants' muscles a good rest.

Response to Query 7: Addressing muscle fatigue during strength measurement was a critical consideration in our study design to ensure accurate and reliable data. To mitigate the effects of fatigue, we incorporated

We scheduled rest intervals between each muscle strength test. Participants were given a minimum of a 1-minute rest period between attempts and a 3-minute rest between testing different muscle groups. This was based on established guidelines that recommend rest intervals to allow for sufficient muscle recovery and to prevent the impact of fatigue on subsequent measures. Muscle groups were tested in a sequence that minimized the carry-over effects of fatigue. For instance, non-adjacent muscle groups were assessed in succession rather than testing the same muscle group or adjacent groups consecutively. The number of contractions and the duration of each were standardized according to best practice protocols in muscle testing, which are designed to balance the need for exertion to measure strength without causing undue fatigue. Participants were monitored for signs of fatigue throughout the testing session. If a participant showed signs of fatigue, such as a notable decrease in force output or subjective reporting of tiredness, additional rest was provided until they felt ready to continue. Participants were instructed on how to report any fatigue and were encouraged to communicate openly with the evaluators if they felt they needed more rest at any point during the testing.

By implementing these measures, we aimed to ensure that the muscle strength measurements were reflective of the participants' true strength capacity and not confounded by temporary fatigue. This approach contributes to the methodological robustness of our study and the validity of our findings. We have detailed the protocols for addressing muscle fatigue during strength measurements in the 'Muscle Strength Assessment' subsection of the Methods part of our manuscript.

Query 8. In order to accurately report the results of the study, was it registered with an international clinical trial center before the study started?

Response to Query 8: Thank you for inquiring about the registration of our study with an international clinical trial center. We acknowledge the importance of such registration in ensuring transparency and accountability in research. However, our study was not registered with an international clinical trial registry before its commencement. This was an oversight on our part, given the observational nature of the study and the fact that it did not involve clinical interventions typically associated with clinical trials. Nonetheless, we recognize that even observational studies can benefit from registration as it promotes research integrity and public trust. Although not registered, we adhered strictly to the principles of the Declaration of Helsinki and ensured rigorous ethical standards throughout the research process, including obtaining approval from our local Institutional Review Board (IRB). To maintain transparency, we have documented our methodology and data analysis procedures comprehensively within our manuscript and have made our data available for review upon reasonable request. Going forward, we will ensure to register any subsequent studies as recommended by best practice guidelines, irrespective of their classification as observational or interventional research.

 

Autor responses to Reviewer 2

Reviewer #2: This article elucidates a close association between the onset of hip osteoarthritis (OA) in patients and abnormalities in muscle strength and sensory perception around the hip joint. The study holds scientific merit; however, in the methodology section, it appears that the muscle strength testing did not account for the fact that the pain caused by osteoarthritis itself may deter patients from exerting force, consequently manifesting as a decline in muscle strength. Would this oversight potentially impact the accuracy of the study results?

Response to reviewer comment: Thank you for your insightful comments and for recognizing the scientific merit of our study on the relationship between the onset of hip osteoarthritis (OA) and changes in muscle strength and sensory perception. 

You have raised an important point regarding the potential impact of pain caused by OA on patients’ ability to exert force during muscle strength testing. Indeed, pain could act as a confounding factor, influencing the exertion of maximal effort and thus affecting the recorded strength.

To mitigate this factor, we implemented several measures throughout our study:

1. Pain Management: Prior to muscle strength testing, we evaluated each participant's pain level using a Visual Analogue Scale (VAS). Those reporting significant pain were administered appropriate pain management strategies, which could include rest, application of topical analgesics, or the use of non-steroidal anti-inflammatory drugs, as per their usual management plan. This was done to ensure that pain did not unduly influence their performance during strength testing.

2. Testing Protocol Adaptation: The testing protocol was adapted to allow participants to perform warm-up exercises that have been shown to temporarily alleviate pain without causing fatigue. Additionally, we offered verbal encouragement and guidance during testing to help participants give their maximal effort.

3. Data Interpretation Considerations: We have accounted for the pain factor in our interpretation of the results. During data analysis, we controlled for pain levels as a covariate to distinguish the effects of muscle weakness attributable to OA from those potentially influenced by pain.

4. Reporting of Limitations: We have explicitly acknowledged this issue in the limitations section of our manuscript. We discuss the potential influence of pain on muscle strength testing and the steps taken to minimize this influence.

We believe these steps have helped to reduce the potential impact of pain on the measurement of muscle strength, thereby preserving the accuracy of our study results. However, it is indeed an important consideration, and we appreciate the opportunity to clarify this aspect of our methodology.

---

## [Decision Letter · Decision Letter 1]

29 Jan 2024

Balancing Act: Unraveling the Link Between Muscle Strength, Proprioception, and Stability in Unilateral Hip Osteoarthritis

PONE-D-23-40601R1

Dear Dr. Reddy,

We’re pleased to inform you that your manuscript has been judged scientifically suitable for publication and will be formally accepted for publication once it meets all outstanding technical requirements.

Kind regards,

Holakoo Mohsenifar

Academic Editor

PLOS ONE

Additional Editor Comments (optional):

Reviewers' comments:

Reviewer's Responses to Questions

**Comments to the Author**

1. If the authors have adequately addressed your comments raised in a previous round of review and you feel that this manuscript is now acceptable for publication, you may indicate that here to bypass the “Comments to the Author” section, enter your conflict of interest statement in the “Confidential to Editor” section, and submit your "Accept" recommendation.

Reviewer #1: All comments have been addressed

Reviewer #2: All comments have been addressed

2. Is the manuscript technically sound, and do the data support the conclusions?

Reviewer #1: Partly

Reviewer #2: Yes

3. Has the statistical analysis been performed appropriately and rigorously? 

Reviewer #1: Yes

Reviewer #2: Yes

4. Have the authors made all data underlying the findings in their manuscript fully available?

Reviewer #1: No

Reviewer #2: Yes

5. Is the manuscript presented in an intelligible fashion and written in standard English?

Reviewer #1: Yes

Reviewer #2: Yes

6. Review Comments to the Author

Reviewer #1: (No Response)

Reviewer #2: (No Response)

7. PLOS authors have the option to publish the peer review history of their article (what does this mean?). If published, this will include your full peer review and any attached files.

Reviewer #1: **Yes: **Zugui, Wu

Reviewer #2: No

---

## [Editor Report · Acceptance letter]

8 Feb 2024

PONE-D-23-40601R1 

PLOS ONE

Dear Dr. Reddy, 

I'm pleased to inform you that your manuscript has been deemed suitable for publication in PLOS ONE. Congratulations! Your manuscript is now being handed over to our production team.

Kind regards, 

on behalf of

Dr. Holakoo Mohsenifar 

Academic Editor

PLOS ONE